# "If diagnosed early, you will be stressed and die…" drivers for breast cancer screening services uptake among women in Dar es Salaam

**Nathanael Sirili[1], Khadija Msami [2], Thadeus Ruwaichi [1] \*, Notikela Nyamle [3]**

**1** Department of Development Studies, Muhimbili University of Health and Allied Sciences, Dar es Salaam, Tanzania, **2** Ocean Road Cancer Institute, Directorate of Cancer Preventive Services, Dar es Salaam, Tanzania, **3** Health and Social Sciences Research Foundation (HSSRF), Dar es Salaam, Tanzania

\* meelathadeus@gmail.com

**Data Availability Statement:** Data have been uploaded as supplementary information.

**Funding:** The study was funded by the Tanzania Comprehensive Cancer Project TCCP 'Breast

## Abstract

Worldwide, there has been an increase in the breast cancer mortality rate, with disproportionately high rates in low and middle-income countries. Addressing breast cancer starts with early detection through screening services. In Tanzania, despite being among countries with high rates of breast cancer, screening services uptake has remained low. This study aimed to explore the drivers for breast cancer screening among women accessing health care services at a specialized cancer treatment hospital in Dar es Salaam, Tanzania. We adopted an exploratory case study employing qualitative techniques to analyze the drivers for breast cancer screening among women in Dar es Salaam. We interviewed four groups of respondents: women with breast cancer attending cancer treatment clinics, young women and old women without breast cancer attending cancer screening units, and older women who do not show up for breast cancer screening. From analysis of the in-depth interviews and focus group discussions we found that the drivers for breast cancer screening operate at different levels; individual as a centre of making the decision to be screened, family as an attribute to decide uptake of screening, the society drivers, the healthcare providers related drivers and health facility related drivers. These attributes were found to influence women's decisions to screen, and the possibility of uptake of breast cancer screening was dependent on family and social motivation. In most instances, women were driven to utilize breast cancer screening when the services were readily available at their neighbouring health facilities. The findings from this study have enlightened that people's decision about utilizing breast cancer screening services is based not only on perceptions of their risk but also on fellow community members who have survived the disease, the experiences of breast cancer screening services from their networks and the performance of healthcare institutions in delivery of such services. The use of breast cancer survivors' support groups to promote breast cancer screening services is advocated from the findings of our study.

Cancer' Research Initiative Grant (TCCP-BCRI-2022-04) to KM. The funders had no role in study design, data collection and analysis, decision to publish, or preparation of the manuscript.

**Competing interests:** The authors have declared that no competing interests exist.

## Introduction

Breast cancer is a public health concern surpassing lung cancer as the most commonly diagnosed cancer in the world [1]. In 2020 an estimate of 2.3 million new cases of breast cancer were diagnosed and 684,996 deaths occurred globally [2]. The distribution of cancer burden is attributed to demographic and epidemiologic changes, although it is exacerbated by increasing Non-Communicable Diseases (NCDs) risk factors associated with globalization and a growing economy [1].

There is an increasing trend of breast cancer mortality in lower income, low-middle-income countries, while in developed countries the mortality, and morbidity trend is decreasing [3]. In Tanzania, breast cancer represents 14.4% of new cancers case among women and the number of new breast cancer cases is projected to increase from 2,732 in 2012, to 4,961 cases in 2030, an increase of 82% [4].

Various breast cancer prevention strategies have been documented in the literature, ranging from primary prevention methods, such as healthy lifestyles, and secondary prevention methods, such as screening and early detection [5]. In Tanzania, self-breast exams and clinical breast examinations by health care professionals are feasible at all levels of the health care system and have been widely used in breast cancer prevention and control campaigns [6]. In contrast to developed countries, the widespread use of advanced breast cancer screening techniques such as mammograms has led to declining breast cancer mortality with robust access to appropriate cancer care [7]. The general recommendation for breast cancer screening is for average risk women to be screened from 50, and from 40 and below only if deemed high-risk [8].

Breast cancer screening uptake is not optimum, as women continue to be diagnosed at advanced stages in Tanzania. The Ocean Road Cancer Institute (ORCI) is a public institution hosting a cancer screening clinic (CSC) which screens women for breast and cervical cancer. While cervical cancer screening is conducted on women from the age of 30, breast cancer screening has no age restriction. In this approach, there is a lot of anxiety associated with screening outcomes, mostly emerging from individual influences, family, community and the healthcare setting driving a person to go in for screening [9].

Younger women are believed to be mainly driven by symptoms to utilize screening services that would have benefited more those at risk for breast cancer. This study set out to elucidate the drivers that influence the screening behavior of different groups of women. In the interest of finding out what exactly could motivate more age-appropriate women to screen, interviews and focus groups were conducted to analyze the drivers for uptake of breast cancer screening services using the social ecological model.

## Materials and method

### Study design

We adopted an exploratory case study using qualitative techniques to analyze the drivers for breast cancer screening among women in Dar es Salaam. A case study approach was deemed appropriate for this study as screening for breast cancer is embedded in the social context of individuals, families and community at large. Therefore, in-depth inquiry is required and thus case study [10].

### Study context

Health care services provision in Tanzania is organized in a pyramid of three levels; primary, secondary and tertiary. The primary level comprises community, dispensaries, health centres,

district hospitals and the council health management teams. The secondary is composed of regional hospitals and regional health management teams and the tertiary is composed of zonal referral hospitals, consultant and national hospitals. The Ministry of Health (MOH) is responsible for overseeing health services provision in the country. Tanzania has three major centres for cancer care to serve a population of 55 million. These centres provide radiation, chemotherapy, and surgery. Ocean Road Cancer Institute (ORCI) is a tertiary specialized hospital for cancer treatment in Tanzania. As the oldest center and the national referral hospital for cancer, ORCI is mandated to offer cancer screening, diagnosis, treatment, palliative care, health education and outreach services. ORCI also has an outpatient medical clinic for non-cancer clients and is a training institution that also engages in research and consultancies with local and foreign parties. Cancer screening services at ORCI cover a diverse patient population from all over Tanzania since cancer screening services are conducted without cost to the client and due to the availability of diagnostic services, are not readily available at other screening centers.

## Organization of cancer services provision in Tanzania

Tanzania's healthcare system faces non-communicable diseases (NCDs) as a major public health threat with cancer becoming increasingly prevalent. The cancer services are primarily provided through a combination of government-run healthcare facilities, private healthcare providers, and non-governmental organizations (NGOs). These institutions, at times in collaboration with international partners, carry out several programs at improving overall cancer services including for screening and prevention. Screening services are offered at facilities from the district level embedded with other services such as reproductive and HIV care and treatment centre. Cancer care in Tanzania, like in many developing countries, faces several challenges. Access to cancer care can be limited in some regions of Tanzania, especially in rural areas where healthcare infrastructure may be inadequate. Although other facilities can have independent oncology departments or operate under larger centres, there still remains a shortage of specialized healthcare professionals, including oncologists, radiologists, and radiation therapists. Funding and resources for cancer care and treatment may be limited, leading to challenges in providing comprehensive services.

## Recruitment of study participants

We purposefully selected the study participants from the cancer-screening clinic (CSC) and at the outpatient clinic. We grouped our participants into two major groups, younger women than the age recommended Breast Cancer Screening (BCS) and older women who fit the age for breast cancer screening. We further subdivided the two groups to women with breast cancer and older women without breast cancer. From the electronic registry at the screening clinic, we identified participants with eligibility to be included in-group I, II and III (Table 1). In total, we recruited and interviewed 12 participants from the three categories. In addition,

**Table 1. Study participants.**

| Group | IDI | FGD |
|---|---|---|
| Young women attending CSC for CBE | 3 | 5 |
| Women with breast cancer | 5 | 5 |
| Older women without cancer. | 4 | 6 |
| Older women who do not show up for screening. | 6 | 0 |
| **Total** | 18 | 16 |

we obtained a group of six older women who have not shown for breast cancer screening at the outpatient department. Furthermore, we conducted three3 FGDs with 16 participants from groups I to 3 (Table 1) across the study populations.

### Recruitment dates

The participants were recruited between Saturday 20[th] of May 2023 to Saturday 17[th] of June 2023.

### Data collection

We used a Kiswahili semi-structured interview and focus group discussion guide to conduct in-depth interviews (IDIs) and focus group discussions (FGDs) with the recruited study participants. The IDIs and FGD guides were developed based on the existing literature on messages regarding BCS messages and knowledge of the researchers on the BCS messages and cancer screening services at ORCI. The questions focused on understanding the drivers for a person to decide to go in for breast cancer screening as an individual, family, community and health care system factors. The question guides aimed to capture nuances in participants' knowledge of breast cancer, the benefits of screening, who is considered eligible to screen and the availability of screening services in their nearby health facilities.

The guides were first developed in English and later translated into Kiswahili before being administered, as most of the study population is fluent in Kiswahili.

Before the actual data collection started, we pre-tested the IDI and FGD guide to a few patients at the outpatient department at ORCI on the second day of research assistants' training to ensure that the guides contained clear questions. After the pre-test, we refined the questions before the actual data collection started.

The in-depth interviews (IDIs) and FGDs were conducted in a designated room close to the screening clinic at ORCI, where participants were comfortable as they were already familiar with the screening clinic environment. A researcher accompanied by a research assistant who recorded the conversation using a digital audio recorder and field notes conducted the IDIs and FGDs. The IDI lasted between 15 and 45 minutes.

For the IDIs, we stopped data collection at the 18[th] interview after attaining information saturation, in each FGD, we stopped the discussion when there was no new information that was coming out of information saturation, where no new information was coming out from conducting more interviews. For the FGDs, we aimed at understanding group dynamics, and we focused on intragroup saturation rather than inter-group saturation. That is, in each FGD, we stopped the discussion when no new information was coming out, even with continuing the discussion.

### Data analysis

We adopted qualitative content analysis as explained by Granheim and Lundman [11]. All the audio transcripts from the IDIs and FGDs were verbatim transcribed before starting the analysis. The lead author guided the authors to revisit the study objectives, the IDI and FGD guides and all together teased out the main domains targeted by the objectives, and data collection guides regarding the breast cancer screening messages. Based on the domains developed a preliminary codebook was developed. The authors then chose a rich transcript and read it manually and individually. After reading individually, the authors came back and jointly discussed issues that were coming out but not reflected in the preliminary codebook and enriched the codebook. With the enriched codebook, the team subdivided into two pairs and jointly coded a similar manuscript. The pairs came together and discussed the codes that they derived from

the transcript, there being no much discrepancy in between the two pairs, coding continued individually to the rest of the transcripts among the group of 4 authors.

After having coded all the transcripts, the team discussed the codes, grouped similar codes, and abstracted them to sub-categories. From the sub-categories, abstraction was used to develop the categories.

The categories and subcategories were presented with succinct quotes. Although the process description sounds linear, it was iterative from the beginning to the end, with each step forward and backwards and a consensus within the team making the central conclusion.

## Ethical considerations

This study obtained ethical clearance from the institutional review board of the ORCI (10/VOL.XXI/123-B). Permission to collect data was obtained from the ORCI management and written informed consent was obtained from each study participant before the interview or focus group discussion. Before the interview or discussion, the participants were assured of anonymity and confidentiality of their identity. Furthermore, all audio records were labeled with anonymous labels that did not give any identifiers of the participants. The audio records were then transmitted to a locked folder in the computer of the main author who shared them to transcribers with anonymity.

## Reliability and validity measures

In this article, we discuss the methodological considerations by discussing both study limitations and trustworthiness.

To enhance trustworthiness, we adopted the four Guba's criteria [12] of credibility, dependability, conformability and transferability. The credibility of these study findings was enhanced through pre-testing of interview and focus group discussion guides, triangulation of study participants, and triangulation of researchers and development of the codebook before analysis. We enhanced the dependability of the study findings through the adoption of a maximum variation sampling strategy in the recruitment of study participants and triangulation of data collection strategies. The conformability of the findings is enhanced through the thick description of the study methodology and the presentation of findings with succinct quotes. Finally, transferability of the findings of this study is enhanced through a thick description of the study context and the methodology used.

## Findings

Analysis of the IDIs and FGDs unveiled four major categories in relation to divers for breast cancer screening (Fig 1). The categories included; (i) individual as a centre towards breast cancer screening uptake attributed to distance to screening services centre, indirect costs, myth on causes of breast cancer, hesitancy to screen, fear of being diagnosed and awareness of breast cancer. drivers, family drivers, community drivers and family drivers. (ii) Family drivers for breast cancer screening uptake attributed support from partners, support from family members, testimony from close relatives and peer influence. (iii) Community-Level Drivers for Breast Cancer Screening attributed to passing messages through religious leaders and messages through political leaders and (iv) Health Facility Drivers for Breast Cancer Screening attributed to perception on the place of service, delayed referral and documentaries on availability of services.

# Drivers for breast cancer screening

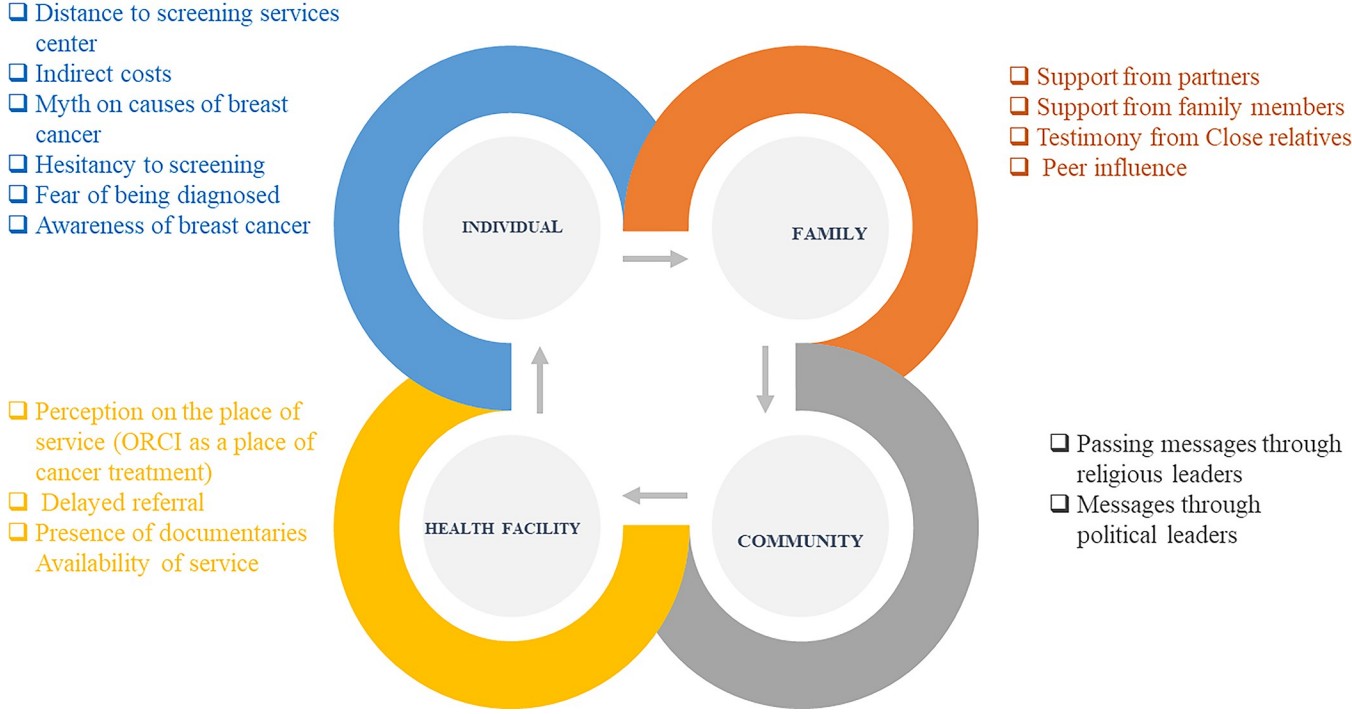

❑ Distance to screening services center
❑ Indirect costs
❑ Myth on causes of breast cancer
❑ Hesitancy to screening
❑ Fear of being diagnosed
❑ Awareness of breast cancer

❑ Support from partners
❑ Support from family members
❑ Testimony from Close relatives
❑ Peer influence

INDIVIDUAL      FAMILY

❑ Perception on the place of service (ORCI as a place of cancer treatment)
❑ Delayed referral
❑ Presence of documentaries
  Availability of service

❑ Passing messages through religious leaders
❑ Messages through political leaders

HEALTH FACILITY      COMMUNITY

**Fig 1. Summary of findings.**

## Individual as a centre of decision towards breast cancer screening uptake

Individual drivers were associated with women going for breast cancer screening. Most women reported several individual drivers, which motivated them to undergo screening. They stated distance from the screening centre, fear of being diagnosed, hidden costs such as time, age factor, myth on the cause of breast cancer, and hesitancy to screening. It is common at first for patients to identify problems and seek to attend a health centre to seek assistance.

## Distance to screening services centre

Participants stated that the distance from home to health facilities drives women to attend health facilities to screen for BC. Some women reported that they failed to attend screening due to the distance between their homes and the health centre. Most reported that the screening services are far from their residents, demanding them to wake up early for successful service.

Thus, distance forces women to travel long and spend much time and effort abandoning their work and families.

"...sometimes leave my home at 4 am to go to the hospital. There was a day when I woke up at 3 am only to get the first appointments in the queue. I waited until 6 am when others started coming. The earlier you come, the earlier you get the first number in the queue... so to get service early, you must come early..." (IDI- older woman-03)

### Indirect costs

The hidden cost was another driver for women to come for screening services or not. They added that such costs are not directly related to the costs of breast cancer screening and they sometimes influence a person to seek screening. Among others, time cost and transport were revealed factors, which may influence someone to attend screening. The time required to attend screening appointments can interfere with individual work. Some of the respondents argued that they attend screening only if they have time and money to cover their transport fares. Respondents explained that women failed to attend screening due to their limited time and sometimes they lack bus fare to go for screening. Respondents knew that the screening is free, but they explained that what makes them not to screen is lack of enough time and costs to cover their transports. Some women explained this as one states that.

"*Cost is another challenge, for example I am living at Mbezi King'azi coming and returning bus fare will cost me 6500 Tsh in total. It costed me a lot the time I was frequently coming for investigations" (FGD 1-older woman member-04*

**Knowledge on breast cancer screening.** Participants stated myths and awareness on the causes of breast cancer that sometimes in society act as another driver for a person to decide to attend screening or not. Some women associate breast cancer with witchcraft and as a disease, which cannot be cured in the hospitals. In the community, there has been a myth that attending hospitals ends up causing people to lose their life instead of cure. In contrast those who were aware of the risks of breast cancer were ready to show up for screening.

"*. . .there are some people who have a negative belief that when someone is suffering from Cancer, they normally say you know she went to see someone? She went to bewitch that family that is why she has also bewitched. . . There are many myths, which I hear from patients. . . "You see how that person is suffering" but when you look at things it is not true that Breast cancer is caused by witchcraft. . ."* (IDI-older woman-04)

They added that these women are curious to know their health status, and thus, it is easier for them to seek breast cancer screening services.

"*. . .for me, I think I am in a better position because it is something which I wished to know so that I won't fall into the breast cancer problem. I sometimes searched Google on the symptoms, causes and preventions; I am a person who likes to get prior information. . ."* (IDI-young woman-03)

### Hesitancy to screening

Existence of screening hesitancy for the women to attend screening in fear of being diagnosed with cancer. Participants added that the majority of the women are not ready to screen as they hesitate, fearing being diagnosed with breast cancer.

"*. . . what I mean is that they say when a day comes, we will also go for breast cancer screening, they also love to be screened, but they hesitate to decide to go for screening. . ."* (IDI-older woman-04)

### Fear of being diagnosed

Some informants stated that they feared being diagnosed with breast cancer. The fear resulted from the fact that women were worried that after they had been diagnosed, they would have to go for treatment, an illness that they sometimes perceived to have no cure. They added that women in the community have the notion that if you are exposed to the truth about your health, you will enter into a state of stress, which may raise your pressure and lead to death, and thus, it is better to stay without being diagnosed to live longer. Thus, to maintain confidence, women thought it was better to delay screening than to be diagnosed early. The respondent explained that their fear in the community of being diagnosed.

> "... you are bringing us tension, so that we go to the hospital to screen, me,I don't have money to attend treatment. If someone has this perception, they will die from stress; You may screen and find that you don't have money when you go home. This will make you die from stress..." (IDI-older woman-03)

On the other hand, lack of awareness of breast cancer screening emerged as a factor related to fear but an independent impediment to women attending the screening. Individuals not knowing the screening process, the causes of cancer, the places where they can attend to get screening and the effects of not being diagnosed at the early stages characterized it. There have been outdated beliefs and myths about the treatment of breast cancer. Some women lacked an understanding of what breast cancer is and its diagnosis.

> "...we had no understanding, no understanding about breast cancer screening; my family didn't know if they were at risk of getting cancer...." (FGD 2-older Woman member-01)

### Family drivers for breast cancer screening uptake

Most of the women in this study reported that different family-level drivers influence them to go for screening. The relationship, which exists at the family level, was found to impact participant screening behaviours and attitudes. The participants reported being influenced by family members. Such influence comes from partners, relatives, and peer influence.

### Support from partners

Participants in this study reported that support from partners was one of the factors influencing women to seek or not seek screening. They explained that there are some influences from the partners/husbands for a woman to show up for screening; there are those partners who encourage their wives to go for screening, and there are those who even escort them. The respondents reported that when they tell their partners about screening, they become happy, encourage them, and give support, like covering all the transport and other costs.

> "... he is supporting me, he does not refuse, he is always happy when I tell him that I am going for screening when I get good results, he is also happy to know that I am not sick, so he always supports me." (IDI-older Woman-01)

On the contrary, there has been a lack of support from partners when women share the news with partners about going for screening. Sometimes the partners become rude and unhappy with the process. Some women are ready to seek the screening but their partners do not support them.

*". . .as number two is saying, there are other men who are stubborn, they must be educated first, you may find I telling my man that I am having pain in my breast, but in few days when you ask permission to go to Hospital he may start asking you what are you going to do, for me I did breast screening. . ..." (FGD 1- Older Woman Member-04)*

### Support from family members

Some informants stated that family members motivate women to attend the screening for breast Cancer. Women get courage and are ready to go for screening when they get support from their family members.

*". . . for me, I told my Mum, and she supported me much with happiness "Go my daughter, it is a good choice", so my mum told me to go and take screening not only for breast cancer and other NCDs . . .." (IDI-young woman-03)*

### Testimony from close relatives

Testimony from close relatives who have attended breast cancer screening was reported, indicating that those who have been screened drive other women to attend Screening. Participants stated that testimonials from Cancer survivors or those who have been screened increase the confidence of women who have not been screened to go for screening.

*".. I decided to go for screening after I got information from my sister who had a sister-in-law who survived Breast Cancer. . .." (IDI-young woman-03)*

### Peer influence

The respondents explained that sometimes, influence from peer members may encourage women to go for screening and, on the other hand, may discourage the screening. Getting support from peers encouraged women who felt they had symptoms to go for screening, and at the same time, the peers increased shock and tension in women who may have perceived the symptoms not to show up for screening.

*". . . I have my neighbour who had the same symptoms, and she has always encouraged me to go for breast cancer screening because my symptoms and her symptoms were similar; she told me to go to Ocean Road so I will get treatments; she used to inspire me frequently. So after her several inspirations, I decided I should come to screen if I am positive or negative. . .(IDI-young woman-01)*

and

*". . .having the symptoms made me curious, one of my friends who was perusing medical studies gave me information about Ocean Road Hospital where cancer screening services are offered. . .I then decided to come. . . (IDI-young woman-03)*

### Community-level drivers for breast cancer screening

The findings from this study revealed community factors influencing women's decision to undergo screening; they stated that different community groups, such as religious and political leaders and peer influence, contributed to the decision to screen.

### Passing messages through religious leader

Participants stated that the messages communicated through religious leaders reach many people and increase the courage of women to attend breast cancer screening. Respondents argued that religious leaders have a great influence on encouraging women to show up for screening.

"... *for instance, I am a believer in a particular religion... I think there should be a time to educate society. If they use their preaching to encourage screening, people will likely attend screening. Religious leaders have been used to encourage testing and screening of other diseases like HIV, TB and people have been showing up for screening (FGD 2-older woman member-04)*

### Messages through political leaders

Political leaders were mentioned to have significant roles in ensuring that women attend screening at the community level. The messages given through leaders in the villages and in the streets are likely to reach the target groups. In the meetings, the leaders can deliver the messages to the community on the BC screening.

"...*for sure local leaders are leading the community... it will be good if announcements are made through these offices. I can remember the (serikali ya mtaa) local leaders at that time they were used to spread the news about screening..." (IDI-older woman-01)*

And

"... *in our area, there are these leaders known as the Shekha in Tanzania mainland you call the wajumbe (representatives), they pass from each street, door to door telling people that ZOP have come and those who have symptoms should go for screening. So, this made many women to show up for screening" (FGD 2-older woman member-05)*

### Health facility drivers for breast cancer screening

Participants stated that facility infrastructures were among the drivers for women for screening. One driver was the misconception/perception that the patients who attend screening or services at ORCI are at critical stages. Other drivers stated include referral from lower-level health facilities, availability of documentaries, and availability of services in nearby hospitals.

### Perception of the place of service (ORCI as a place of cancer treatment)

Participants stated that ORCI is considered a dangerous place. Women believe that when someone with breast cancer attends this place, the breast will be taken off instead, so women put a negative perception in their minds of Ocean Road as a place with only sick people. In the community, when people hear that someone is attending ORCI for screening or clinic, they are afraid that Ocean Road is only for those who are critical.

"... *for instance, on the first day when I was told that I had to go to Ocean Road cancer hospital..., When I told my sister that I was going for screening at Ocean Road, she was shocked... he!.... what are you going to do there?...." (FGD 1-older Woman member-04)*

and

". . .there are those who are only afraid due to the Ocean Road Cancer Institute name itself, the name itself increases someone's fear, but if you only bring a tent here, the fear will not be there. . ." (IDI-non screened Woman-04)

## Delayed referral

Informants stated that delays in being referred to tertiary hospitals when a patient is suspected of breast cancer symptoms. The referral process takes long, especially for those who start from the lower lever health centres. Respondents explained that they are sometimes discouraged from attending screening due to the process of being referred to higher levels for screening.

". . .I have attended many hospitals, and from these hospitals, I am given medicines, but I don't see any recovery until when I went to a public hospital which referred me to a specialized Hospital" (FGD 1-older women Member-06)

## Presence of documentaries

Some women stated they are motivated to attend screenings when they see documentaries in different health centres showing women who are screened. They also mentioned that for other women to be encouraged, many documentaries should be displayed in the health centres showing the whole process of breast cancer screening and some patients who are witnesses/ survivors of breast cancer.

## Availability of service

Informants stated that the availability of services near the community is one of the driving factors for women to undergo screening. They added that the closer the services are provided to the community, the more they are ready for screening. The findings of this study revealed that women did not know the places to go for breast cancer screening, but they claimed that if the services are brought to their health centres, they will be attracted to go for screening.

"For instance, there are these hospitals here in Mbezi, and we have another one in Kwembe and the nearby Mloganzila. If those hospitals could have these services, it could have been easy for people here to show up for screening instead of going far for screening. . ." (IDI-older woman-03)

*Proximity of screening services.* Participants stated that services closer to people increase the chances of being utilized. They explained that when screening services are brought closer to people, several women show up for screening.

". . .for what you have said about tents, I think you should bring tents; many people will show up for screening, I tell you, people will come running, even those who do not have money who see themselves as poor will come, and sometimes they will leave their activities only for screening. . ." (IDI-non-screened woman-06)

## Discussion

We aimed to explore the drivers for breast cancer screening among women accessing health care services at a specialized cancer treatment hospital in Dar es Salaam, Tanzania. Individual drivers have been found to have a great influence on women to show up for screening in general and breast cancer in particular [13]. Specifically, factors such as the distance from the health facility where the screening service is available to the area where an individual is living, women are ready to go for screening only if the services are found in their nearby places. Women were likely to attend breast cancer screening in nearby places to reduce the hidden costs of transport and time. This is similar to the studies conducted in Nigeria and Chicago on sociodemographic characteristics, distance to the clinic, and breast cancer screening results which found that differences in spatial accessibility was among the driver for women's choices to attend screening and clinics [14,15].

In addition, some women did not show up for screening due to a lack of understanding and awareness of the breast cancer causes, symptoms, and the importance of going for screening. It was only if the women could have the clear knowledge on breast cancer screening, they were ready to go for screening. This lack of knowledge on breast cancer symptoms has been previously described in women from Northern Tanzania [16]. Similarly, a study on barriers and facilitators on breast cancer screening in Victoria revealed the same issues that women prefer not to know if they had bowel or breast cancer as they did not want their lives to be affected by the diagnosis and subsequent treatment this was due to lack of knowledge and awareness [17]. Messages on breast cancer screening need to be designed to increase the awareness of women on why they should go for screening. This suggests a need to revisit breast cancer screening messages to include details that are more educational and strengthen tailored awareness campaigns to dispel the myths and negative perceptions hence potential to increase uptake of breast cancer screening.

On the issue of cost, other studies conducted on early detection of breast cancer in countries with limited resources found that organized screening has a great motivation for women to screen for breast cancer as this helps to cut costs [18]. Tanzania implements an opportunistic screening strategy with periodic mass screening campaigns that mainly rely on healthcare personnel to perform the clinical breast exam, as such, the women studied did not make remarks of high direct costs of care (transactional costs) instead the women were burdened with opportunity costs of travel and time. It is notable, however, that the reimbursement schemes for health delivery in Tanzania favor treatment services over preventative services, including cancer screening [19]. This seems to be commonplace in Africa as seen in Kenya, Rwanda, and Ghana presenting a lost opportunity for driving uptake in screening. Kenya has shown a similar burden of cost related to travel and time for services, particularly women without breast cancer that can lead to late-stage diagnosis. The mitigator proposed was to have more screening clinics in a wider geographic setting. Similar to our setting, in Kenya, the costs for screening are primarily offset by policy.

On the other hand, family drivers were found to influence women in making decisions of whether to screen or not, the fear increased when women think of being diagnosed and revealing it to family members. Family members' motivation and support played a great role. In this study, the women who had support from partners were likely to go for screening compared to those whose partners did not give support. Motivation from a close relative or family increased the health seeking behavior to a person compared to those who do not get support, in this case, many fears arise including stigma and discrimination when someone is diagnosed positive and it makes women not to go for screening, as they were afraid of stigmatization in the family and community. Those who had witness survivors or who received messages from witness

survivors were ready to screen. The findings of this study goes in line with the findings of the study conducted in South North Carolina on HIV Testing Behaviors of Young Black College Women which found that interpersonal relationships have a great impact on women in health seeking behavior and attitude to show up for testing [20]. In Addition, lack of encouragement by family members and physicians is among the major inhibitors to women's participation in breast cancer screening [21]. Interestingly, a study from women in Kenya found that symptoms were a driver for screening and that the women are complacent up to when they encounter a "problem" is when they seek health care [22].

In this study, community drivers were found to be messages from local leaders, religious leaders and political leaders who had great influence in the community. In the community, people tend to have trust in the people who lead them and those who they interact with most. A message given by religious leader is likely to be trusted by their followers compare to those given from other people who are announcing screening. Likewise, the local leaders are there always to solve the problems in the community therefore receiving a message to go for screening from a local leader it is likely to be accepted compare to a message received from other sources. The study in Asia on barriers for breast cancer screening among Women supports the findings of this study that health education through use of media and community health programs created awareness of earlier presentation and diagnosis of breast cancer in Asian women and motivated the participation in breast cancer screening programs [21].

Stigma that stems from embarrassment is one social barrier to breast cancer screening that did not come up in the discussions with these women which is contrary to some literature on the subject [23]. However, even in other areas with lower reporting of stigma, this is still a threat to screening for certain minority backgrounds [24]. This finding of considerably less weight placed on embarrassment as a barrier to screening was similar to other Tanzanian women [16]. Suggesting that the other barriers to screening need more attention.

Lastly Facility drivers were also found to have great influence on women to go for screening, from the finding the barrier for women not to show up for screening was a perception of a facility, the facility was perceived to be a place for only those who are seriously sick. The facility made people to avoid/not to show up due to lack of knowledge and understanding of the Breast cancer screening. These findings go in line with a study conducted in Malawi though it may differ due to the issues being studied. But in terms of health seeking behavior they concur, the study found the similar perception on health seeking where as women did not see the reason for going to health facilities for screening when they are healthy, that is, they do not have any symptoms [25].

The findings of this study bring to light several policy implications for promoting breast cancer screening. Firstly, tailored messages to the diverse age groups are needed for increasing screening services uptake. Tailoring should focus on the content and media to which the messages are delivered. Secondly, promoting testimonies from breast cancer survivors (peer to peer) has the potential to increase the uptake of breast cancer screening. Thirdly, health promotion campaigns through radio, television and other public media, including community gatherings, have the potential to raise awareness of breast cancer and the benefits of breast cancer screening. Lastly, scaling up screening services to other facilities than ORCI to include outreach services is key in breaking the barriers towards access to screening services.

## Study's limitations

The fact that this study was conducted in the hospital setting may limit reaching a wider scope of participants to capture a broader content regarding drivers for breast cancer screening.

However, inclusion of the diverse groups of women adds strengths to this study; also, the fact that these women came from different geographical locations offset the limitation.

## Conclusion and recommendations

In this study, drivers related to breast cancer screening uptake have been analyzed using the SEM. Drivers at all levels of the SEM impact having gotten a screening test. Notably, older women tend to be less knowledgeable about the disease and their risk for it, therefore less likely to know about treatment options and the opportunity for survival given early detection. The fear within individuals of this group of women, coupled with the interpersonal and institutional levels, have significant effects on breast cancer screening uptake.

### Area for further research

Further research is recommended on exploring context-specific strategies for increasing the uptake of Breast cancer screening among women in different contexts. Furthermore, research on financial support programs (e.g., screening reimbursement) for breast cancer screening is recommended.

## Supporting information

**S1 Data.**
(ZIP)

## Acknowledgments

We would like to thank the women who agreed to be interviewed and offered so much insight into our questions and ORCI staff for their support. All authorities which gave us permission to conduct this study are acknowledged.

## Author Contributions

**Conceptualization:** Nathanael Sirili, Khadija Msami.

**Data curation:** Nathanael Sirili, Khadija Msami, Notikela Nyamle.

**Formal analysis:** Nathanael Sirili, Khadija Msami, Notikela Nyamle.

**Funding acquisition:** Khadija Msami.

**Investigation:** Nathanael Sirili, Khadija Msami.

**Methodology:** Nathanael Sirili, Khadija Msami.

**Project administration:** Khadija Msami.

**Supervision:** Nathanael Sirili, Khadija Msami.

**Validation:** Nathanael Sirili, Khadija Msami, Thadeus Ruwaichi.

**Visualization:** Khadija Msami.

**Writing – original draft:** Nathanael Sirili, Khadija Msami, Thadeus Ruwaichi, Notikela Nyamle.

**Writing – review & editing:** Nathanael Sirili, Khadija Msami, Thadeus Ruwaichi, Notikela Nyamle.

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
