## [Decision Letter · Decision Letter 0]

1 Aug 2024

PGPH-D-24-01232

“If diagnosed early, you will be stressed and die…” Drivers for uptake of breast cancer screening services among women in Dar es Salaam.

Dear Dr. Ruwaichi,

Thank you for submitting your manuscript to PLOS Global Public Health. After careful consideration, we feel that it has merit but does not fully meet PLOS Global Public Health’s publication criteria as it currently stands. Therefore, we invite you to submit a revised version of the manuscript that addresses the points raised during the review process.

EDITOR's comments to the authors: 

Ensure that the manuscript is revised in accordance with the author's guidelines and that the appropriate checklist is completed (see https://journals.plos.org/globalpublichealth/s/submission-guidelines#loc-materials-and-methods)Data Collection Methods - The author should provide more information about some of the specific questions listed in the question guide. The subtitle 'methodological consideration' in the material and methods section should be revised to reliability and validity measures.Results- Ensuring that the study's results are clearly presented and linked back to the research objectives will help readers understand the significance of the findings. A well-structured presentation of findings can help the research have a greater impact. As an example, there is a disconnect on how individual-level factors relate to indirects cost, distance to clinic.... You can keep it as it is but provide a linking narratives for the readers.  Also, awareness of SBE or CBE and myths can each be under  a sub-category - knowledge. Overall, for the results section, provide proper narrative linking the emerged themes to the figure 1 designed and how domains in the SEM model was covered. Before the conclusion and recommendation section, It would be beneficial to include a section discussing the study's limitations. Addressing potential biases, data collection limitations, or other factors that may have influenced the results will improve the overall research.Recommendations for Future Research: Identifying areas for future research on breast cancer screening drivers in various contexts or populations could help to deepen the study. Proposing new avenues of research can help to advance the field of study.Lastly, consider grammar editing to make sure there are no errors and provide an English translated transcripts. 

We look forward to receiving your revised manuscript.

Kind regards,

Kelechi Elizabeth Oladimeji

Guest Editor

Journal Requirements:

Additional Editor Comments (if provided):

Reviewers' comments:

Reviewer's Responses to Questions

**Comments to the Author**

1. Does this manuscript meet PLOS Global Public Health’s publication criteria? Is the manuscript technically sound, and do the data support the conclusions? The manuscript must describe methodologically and ethically rigorous research with conclusions that are appropriately drawn based on the data presented.

Reviewer #1: Yes

Reviewer #2: Yes

2. Has the statistical analysis been performed appropriately and rigorously?

Reviewer #1: N/A

Reviewer #2: N/A

3. Have the authors made all data underlying the findings in their manuscript fully available (please refer to the Data Availability Statement at the start of the manuscript PDF file)?

Reviewer #1: Yes

Reviewer #2: Yes

4. Is the manuscript presented in an intelligible fashion and written in standard English?

Reviewer #1: Yes

Reviewer #2: Yes

5. Review Comments to the Author

Reviewer #1: It's an important topic and the researcher tried to evaluate the main concern about the reason of not participate in screening. This is a small scale study, need broad scale study for more details and find out the concrete cause for implementation of intervention. Can be published.

Reviewer #2: PGPH-D-24-01232

“If diagnosed early, you will be stressed and die…” Drivers for uptake of breast cancer screening services among women in Dar es Salaam

The study presented a well-structured and informative study on the drivers influencing women in Tanzania to seek breast cancer screening. The study used a qualitative approach with clear justification and detailed methodology. The interview guides were thorough and the data collection procedures ensured consistency and trustworthiness. The discussion provided strong evidence by effectively connecting findings to existing research, highlighting the relevant literature. Finally, the study pinpoints areas where the results differ from previous research, suggesting avenues for further exploration.

However, authors should discuss policy implications based on the study findings. Example, this study revealed that many women lacked awareness about breast cancer, its symptoms, the importance of screening, and the benefits of its early detection. What strategies can address these knowledge gaps? (You may consider educational campaigns utilizing various channels such as radio, television, community meetings, and educational materials in the local language to reach a wider audience). Additionally, the study reported that the financial burden (transportation and time off work) emerged as a significant barrier to screening. What role can reimbursement programs play in offsetting these costs, making screening more accessible to women, especially those from lower socioeconomic backgrounds?

6. PLOS authors have the option to publish the peer review history of their article (what does this mean?). If published, this will include your full peer review and any attached files.

**Do you want your identity to be public for this peer review?** For information about this choice, including consent withdrawal, please see our Privacy Policy.

Reviewer #1: No

Reviewer #2: No

---

## [Editor Report · Decision Letter 1]

13 Sep 2024

“If diagnosed early, you will be stressed and die…” Drivers for breast cancer screening services uptake among women in Dar es Salaam.

PGPH-D-24-01232R1

Dear Dr Ruwaichi,

We are pleased to inform you that your manuscript '“If diagnosed early, you will be stressed and die…” Drivers for breast cancer screening services uptake among women in Dar es Salaam.' has been provisionally accepted for publication in PLOS Global Public Health.

In the interim, you need to attend to the following as they seem missing from the your revised submission.

1. A response to reviewer comments for the revised submission made, I could not see that document.

2. Transcript/data uploaded is still in the local language and I did not see the version translated to English version.

3. The Figure in the result section needs to first be interpreted or narrated and cited in the results section to ensure that it is properly linked to other sections of the results section.

Best regards,

Kelechi Elizabeth Oladimeji

Guest Editor
